# Blood Transfusion for Major Trauma in Emergency Department

**DOI:** 10.3390/diagnostics14070708

**Published:** 2024-03-27

**Authors:** Angela Saviano, Cesare Perotti, Christian Zanza, Yaroslava Longhitano, Veronica Ojetti, Francesco Franceschi, Abdelouahab Bellou, Andrea Piccioni, Eugenio Jannelli, Iride Francesca Ceresa, Gabriele Savioli

**Affiliations:** 1Department of Emergency Medicine, Fondazione Policlinico Universitario A. Gemelli, IRCCS, Largo A. Gemelli 8, 00168 Rome, Italy; angela.saviano@policlinicogemelli.it (A.S.); francesco.franceschi@unicatt.it (F.F.); 2Division of Immunohaematology and Transfusion, Fondazione IRCCS Policlinico San Matteo, 27100 Pavia, Italy; c.perotti@smatteo.pv.it; 3Geriatric Medicine Residency Program, University of Rome “Tor Vergata”, 00133 Rome, Italy; christian.zanza@live.it; 4Department of Anesthesiology and Perioperative Medicine, University of Pittsburgh, Pittsburgh, PA 15260, USA; lon.yaro@gmail.com; 5Department of Emergency Medicine-Emergency Medicine Residency Program, Humanitas University-Research Hospital, 20089 Rozzano, Italy; 6Università Cattolica, 00168 Roma, Italy; veronica.ojetti@unicatt.it (V.O.); andrea.piccioni@policlinicogemelli.it (A.P.); 7Department of Emergency Medicine, Wayne State University School of Medicine, Detroit, MI 48201, USA; abellou402@gmail.com; 8Institute of Sciences in Emergency Medicine, Department of Emergency Medicine, Guangdong Provincial People’s Hospital (Guangdong Academy of Medical Sciences), Southern Medical University, Guangzhou 510080, China; 9Department of Orthopedics and Traumatology, Fondazione Policlinico San Matteo, 27100 Pavia, Italy; eugenio.jannelli@libero.it; 10Department of Emergency Medicine, Humanitas University, 20089 Milan, Italy; irideceresa@gmail.com; 11Department of Emergency Medicine, Fondazione Policlinico San Matteo, 27100 Pavia, Italy

**Keywords:** major trauma, bleeding, emergency department, blood transfusion, coagulation

## Abstract

Severe bleeding is the leading cause of death in patients with major trauma admitted to the emergency department. It is estimated that about 50% of deaths happen within a few minutes of the traumatic event due to massive hemorrhage; 30% of deaths are related to neurological dysfunction and typically happen within two days of trauma; and approximately 20% of patients died of multiorgan failure and sepsis within days to weeks of the traumatic event. Over the past ten years, there has been an increased understanding of the underlying mechanisms and pathophysiology associated with traumatic bleeding leading to improved management measures. Traumatic events cause significant tissue damage, with the potential for severe blood loss and the release of cytokines and hormones. They are responsible for systemic inflammation, activation of fibrinolysis pathways, and consumption of coagulation factors. As the final results of this (more complex in real life) cascade, patients can develop tissue hypoxia, acidosis, hypothermia, and severe coagulopathy, resulting in a rapid deterioration of general conditions with a high risk of mortality. Prompt and appropriate management of massive bleeding and coagulopathy in patients with trauma remains a significant challenge for emergency physicians in their daily clinical practice. Our review aims to explore literature studies providing evidence on the treatment of hemorrhage with blood support in patients with trauma admitted to the Emergency Department with a high risk of death. Advances in blood transfusion protocols, along with improvements in other resuscitation strategies, have become one of the most important issues to face and a key topic of recent clinical research in this field.

## 1. Introduction

Major trauma is a key global public health issue and an important financial burden for hospitals and society [1]. Traumatic events are responsible for more than 7–8 million worldwide deaths annually, representing 8% of all deaths [2]. The World Health Organization reported an increase in car and motorcycle accidents, falls, episodes of violence, and other traumatic injuries (involving guns, weapons, and firearms) leading to patient death [3]. Post-traumatic massive bleeding and uncontrolled hemorrhagic shock remain the two major life-threatening events to face and manage to improve patient outcomes and save lives [4]. Studies indicate that 50% of deaths occur within minutes of a traumatic event as a result of massive hemorrhage, 30% are related to neurological dysfunction, and the remaining 20% of deaths occur due to multiorgan failure and sepsis [5]. Evidence agrees with the idea that some deaths related to traumatic bleeding events are preventable if patients receive timely identification of bleeding sources and prompt blood support measures [6]. In the last decade, different blood transfusion protocols and practice guidelines for managing hemorrhage have been developed. These attempts have set up standardized measures to minimize traumatic blood loss, achieve hemodynamic stability, restore tissue perfusion, and reduce short- and long-term organ damage [7]. It is important to know that patients undergoing blood transfusions may experience acute and chronic complications such as allergic reactions, febrile non-hemolytic transfusion reactions, transfusion-related acute lung injury, transfusion-associated graft-versus-host disease, transfusion-associated circulatory overload, hemolytic transfusion, alloimmunization, transfusion-related immunomodulation, and iron overload [8]. Equally important is the risk of disseminated intravascular coagulation (DIC) due to coagulation abnormalities resulting from massive blood transfusion, which can increase the risk of death [9]. Despite ongoing advances in understanding these complex mechanisms, some questions remain unanswered.

## 2. Materials and Methods

This narrative review included papers published in the last twenty years on the use of blood transfusions in patients with major trauma admitted to the Emergency Department. We searched for literature reviews, observational studies (cross-sectional, case-control studies, etc.), retrospective and prospective studies, and clinical trials. We extracted data based on the period of research, title, abstract, and study type. No language restrictions were applied. We searched Up-to-Date^®^, PubMed^®^, Web of Science^®^, and Cochrane^®^. Ethical approval was not required for this review. The principal words we searched were as follows: blood transfusion AND trauma AND Emergency Department; blood components AND hemorrhage AND/OR trauma; bleeding AND blood transfusions protocols AND/OR Emergency Department; pre-hospital setting AND/OR traumatic event AND/OR blood transfusions; and critical care AND/OR traumatic bleeding AND/OR blood transfusions. To summarize, our research question based on PICOST includes the following: population—adults patients admitted to the Emergency Department after traumatic injury with severe bleeding; intervention—massive transfusions, blood products; comparison—blood transfusions and other blood products (platelets, plasma, cryoprecipitate); outcomes—patients survival at 24 h, 48 h, 30 days from the admission to the Emergency Department using blood products with different ratio of red blood cells, plasma and platelets; study type—observational studies (cross-sectional, case-control studies, etc.), retrospective and prospective studies, and clinical trials.

## 3. Definition of Major Trauma

Major trauma has been defined as a “significant injury or injuries that have the potential to be life-threatening or life-changing sustained from either high or low energy mechanisms, especially in those rendered vulnerable by extremes of age” [10]. Approximately one-third of patients with severe trauma admitted to the hospital develop coagulopathy with a high life-threatening risk. It has been reported that adolescents aged 10–25 years are often involved in road accidents, self-harm injuries, and interpersonal episodes of violence as a form of major trauma [1]; from 26 to 50 years, road injuries are mainly responsible for major trauma and post-traumatic bleeding [11] (Figure 1). Another definition of major trauma is based on a threshold Injury Severity Score (ISS)  ≥  15, which is derived from the Abbreviated Injury Scale (AIS) revised in 1998, 2008, and 2015 [12]. Other scores include the New Injury Severity Score (NISS), Revised Trauma Score (RTS), and Trauma and Injury Severity Score (TRISS) [13]. 

The ISS is a standardized tool that was initially applied to patients with blunt trauma due to motor vehicle accidents [14]. This was calculated using a combination of six points (1. minor, 2. moderate, 3. serious, 4. severe, 5. critical, and 6. maximum/currently untreatable) administered to six body regions: (1. head or neck, including the cervical spine; 2. face, including the facial skeleton, nose, mouth, eyes, and ears; 3. chest, including the thoracic spine and diaphragm; 4. Abdominal and pelvic contents: abdominal organs and lumbar spine; 5. Extremities or pelvic girdle: pelvic skeleton; 6. External). The most severe injury from each of these six body systems is assigned an AIS score on a scale from 0 (no injury) to 6 (non-survivable injury) [14]. The ISS scores range from 1 to 75 (ISS = A2 + B2 + C2, where A, B, and C are the squares of the AIS scores of the three most injured ISS body regions) [15].

## 4. Physiopathology of Coagulopathy and Bleeding in Trauma

Coagulopathy associated with trauma includes hypocoagulability in the early hours, which typically results in bleeding and hypercoagulopathy that is associated with venous thromboembolism and/or multiple organ failure [16]. 

Kushimoto et al. [17] defined acute traumatic coagulopathy (ATC) as an endogenous impairment of hemostasis that occurs early after injury. Laboratory tests may show a 50% prolongation of prothrombin time (PT) or partial thromboplastin time (PTT). A PT ratio > 1.2 is associated with worse outcomes. This value is derived from the correlation between clotting times and the incidence of dilutional microvascular bleeding in injured patients who received massive transfusions [18]. Trauma-induced coagulopathy (TIC) is caused by severe tissue damage with subsequent hemorrhagic shock and inability to form clots, leading to excessive bleeding. An INR > 1.5 has been shown to be associated with an increased risk for adverse outcomes in a subset of trauma patients [19] (even if INR is not an accurate predictor of coagulopathy of trauma, and it overestimates the coagulopathy in stable trauma). On the other hand, INR > 4.5 increases the risk for a major bleed [19]. Furthermore, patients can develop platelet dysfunction and hyperfibrinolysis, which is a dangerous cycle [19].

Many pathophysiological mechanisms have been proposed to explain post-traumatic coagulopathy (Figure 2) [1,16,17,18]. First, tissue damage and the state of shock are responsible for endothelial activation, immune system stimulation, platelet and clot activation, and other events that can lead to a ‘lethal triad’ (coagulopathy, acidosis, and hypothermia associated with poor outcomes up to death). After a traumatic event, coagulopathy can switch from a hypocoagulation phenotype to hypercoagulation (depending on tissue damage, shock, and time of injury) [16]. The risk of DIC is particularly high. Three main pathophysiological mechanisms have been described. The first is coagulation activation, hyperfibrinogenolysis, and consumption of coagulation factors [20]. The lethal triad (coagulopathy, metabolic acidosis, and hypothermia) was initially considered a life-threatening effect of post-traumatic bleeding [21]. Patients with post-traumatic bleeding can develop hemorrhagic shock due to blood volume depletion and subsequent reduction in oxygen delivery to the microvessels and microcirculation, leading to a state of metabolic acidosis [22]. Isolated and transient hemorrhagic shock may be tolerated [23]. Hypocoagulopathy derived from the replacement of blood loss with large volumes of fluids and blood supports that lead to dilution of proteins and enzymes required for the formation of clots [24]. Tissue damage from endothelial dysfunction can activate the coagulation system (typically associated with the severity and extent of tissue injuries) [25]. The endothelial cells regulate coagulation, inflammation, and microcirculation. Microvessels represent a crucial barrier for vascular homeostasis and oxygen transport [26]. Post-traumatic endotheliopathy is a phenomenon characterized by the loss of barrier function, adhesion of white blood cells (leukocytes), and the activation of endothelial cells, with manifestations of microbleeding, microthrombosis, and even multiple organ dysfunction [26]. An unbalanced state among components of clots, for example, an insufficient concentration of thrombin, can lead to diminished stability and fibrinolysis [27,28]. Fibrinogen is the most abundant factor in blood, with circulating blood levels of approximately 2–4 g/L in healthy adults [29]. It plays an essential role in clot formation and provides stability. In major trauma, many factors have been identified as causes of fibrinogen depletion, such as blood loss, dilution due to fluid support, impaired platelet and clot function, hyperfibrinolysis, acidosis, and hypothermia [29]. Studies have reported that low fibrinogen concentration is associated with an increased risk of vascular bleeding. Blood tests to measure blood fibrinogen concentration are recommended for injured patients admitted to the ED. The early substitution of fibrinogen, if low, has been associated with better outcomes [27,30].

## 5. Brief Discovery of Blood Transfusion in Trauma

In the pre-modern era, the risks and benefits of blood transfusion in patients with hemorrhage were unknown. In the 17th century, an English physician, Dr. Lower, provide potential support for the treatment of bleeding during blood transfusion [31]. In animals, the benefits of blood support in the case of vascular surgery have been demonstrated. Blood transfusions were also used during hemorrhage after childbirth [31]. The modern transfusion era started in 1901 with the discovery of human blood groups A, B, and O. Subsequently, some advancements have been made in the storage, preservation, collection, and distribution of blood. During World War I, resuscitative measures with blood transfusions gained popularity among patients who suffered from traumatic events. The Coombs test opened a new era of transfusion. After 1960, plastic bags were invented facilitating the conservation and use of blood therapy. Blood transfusions have become safer over the years, and they have been checked for hepatitis B, hepatitis C, virus HIV-1, virus HIV-2, and *Treponema pallidum* [31] and are much more commonly used in patients with post-traumatic bleeding. Currently, transfusion of blood products has become more common and safer [32]. Red blood cell unit were transfused to correct oxygen delivery in patients with anemia. Plasma compounds can clinically improve coagulopathy in patients at high risk of bleeding. Evidence underlines that abnormal laboratory tests of coagulation (PT and INR) are not predictive of clinical bleeding and should not be corrected with transfusions of plasma [33]. Platelets are used to prevent and treat bleeding in cases of thrombocytopenia and other platelet dysfunctions [34]. Some adverse reactions can occur during or after the transfusion of blood products. They mainly include volume and circulatory overload or immune reactions to blood. Nonetheless, blood banks have made efforts to create safer products and support hemovigilance [35], (Figure 3).

## 6. Definition of Massive Blood Transfusion and Current Protocols 

Severe bleeding or massive blood loss is associated with a high mortality risk. Massive transfusion protocol (MTP) regulates the process of blood transfusions that are crucial for its treatment. A study by Patil et al. [36], based on medical literature researches, defined massive blood transfusion (MBT) as: 1. the restoration of one entire blood volume within 24 h; 2. the transfusion of more than 10 units of red blood cells in 24 h; 3. the transfusion of more than 20 units of red blood cells in 24 h [36]; 4. the transfusion of more than 4 units of red blood cells in 1 h when it is needed; and 5. the restoration of 50% of the total blood volume within 3 h. A definition for daily clinical practice is missing, and the remaining definitions are sometimes not useful during a brief period of blood loss as they are based on 24 h intervals. In the past, the management of severe hemorrhage followed the “Berne” concept, which involved multiple-step resuscitation measures: first, with red blood cells, then (if the patient survived long enough) with plasma (up to one blood volume), and finally with platelets (when two blood volumes were required). The administration of higher doses of platelets, plasma, cryoprecipitate has been associated with improved outcomes only if given early [37]. Military services prefer whole blood, including yellow products (platelets, plasma, cryoprecipitate) in cases of post-traumatic severe bleeding. 

A military post-study in Iraq started suggesting the use of fresh, frozen plasma, red blood cells, and platelets for resuscitation. After some years, it was established that coagulation tests (as prothrombin level and partial thromboplastin time) were needed to identify who needed plasma-support. However, these tests showed some clinical limitations in daily routine, especially for patients with massive post-traumatic bleeding because they can initially show normal results, leading to the development of new protocols. Beyond this, damage control resuscitation measures are essential within the first hour (golden hour) or even better within 15 min. In 2006, the Baltimore trauma group [38] stated that first step included damage control resuscitation measures, and then the prevention of post-traumatic coagulopathy was superior to treating it. The authors recommended the balanced ratio of 1:1:1 (red blood cells/plasma/platelets) as easy to use in the first phase. Additionally, the authors observed that higher doses of plasma and platelets were associated with improved outcomes. 

Over the years, to better predict the need for transfusions and the risk of trauma-induced coagulopathy, Yucel et al. [39] analyzed the clinical and laboratory data of patients who underwent massive transfusions. They supported the idea that the use of a score could be useful in the emergency setting. Authors observed the efficacy of the Trauma-Associated Severe Hemorrhage (TASH) Score in predicting the probability of massive transfusions (>10 units of red blood cells) after a traumatic event, directly in the context of emergency or intensive care units [39]. The TASH Score helps identify patients who should receive massive blood transfusions after a traumatic event. The TASH Score includes heart rate, sex, systolic blood pressure, hemoglobin, focused assessment for the sonography of trauma (FAST), base excess, and pelvic/extremity fractures. The score ranged from 0 to 28, where each point represents a percent risk of massive transfusion. For example, a TASH Score ≥ 16 points indicates a probability of massive transfusion > 50%. The TASH Score was considered the best available score by authors [40].

In 2009, Nunez et al. [41] identified the accuracy of another score, the ABC Score (Assessment of Blood Consumption Score) that used simple and non-laboratory parameters (penetrating mechanism, systolic bloop pressure < 90 in the emergency department, heart rate > 120, positive FAST) to prompt understanding of the need of patients in requiring massive transfusions after trauma [41]. In the ABC Score [42] studies, a patient with a score point < 2 was unlikely to require a massive transfusion (sensitivity estimated at 75–90% and specificity estimated at 65–85%, respectively). 

In 2018, Schroll et al. [43] analyzed data from 645 patients and found that both the ABC Score and shock index were useful in predicting the need for massive transfusions after a trauma. The shock index (calculated as the heart rate divided by systolic blood pressure) was more sensitive and required fewer technical skills compared to the ABC score [41]. 

The Modified Early Warning Score (MEWS) and Circulation, Respiration, Abdomen, Motor, and Speech (CRAMS) Score are useful in predicting trauma severity and mortality for multiple trauma-hospitalized patients. The CRAMS Score was found to be superior to MEWS in predicting trauma severity and aiding in making rapid decisions in the emergency setting [44].

Pommerening et al. [45], analyzing data from 1245 patients, confirmed that some scores could be useful in the emergency setting but they pointed out that scores needed to be implemented with an accepted algorithm that considered many other risk factors and parameters in order to better identify patients requiring massive blood support in case of trauma. In fact, 221 (23%) out of 1245 (who received blood transfusions) based only on the score should not have undergone massive transfusions. Guerado et al. [46] stated that some tests (hemoglobin, base deficit, serum lactate, coagulation tests, fibrinogen, and platelets) were sensitive markers for detecting the severity of bleeding and monitoring it in patients who require prompt blood component support [47]. In 2013, Callcut et al. [48] in the Prospective Observational Multicenter Major Trauma Transfusion (PROMMTT) study evaluated the data of about 1245 prospectively enrolled patients. A total of 297 patients received blood massive transfusions. The authors analyzed systolic blood pressure, heart rate, temperature, the international normalized ratio (INR), positive results for focused assessment for the sonography of trauma examination, hemoglobin, base deficit, and penetrating injury mechanism. They concluded that the transfusions of at least three units of blood products with higher plasma and platelet ratios (administered early) were associated with a decreased mortality. They considered the first 24 h after ED admission. A higher plasma and platelet ratios did not correlate with 30 days survival [46,47,48,49]. These findings were confirmed by Holcomb et al. [50] who conducted a prospective study, observing that in the first 6 h after a trauma, injured patients who received a plasma/platelets ratio of 1:1 or higher (in addition to blood transfusions) had a decreased mortality. After 24 h, plasma and platelet ratios were not associated with mortality [50]. An increased plasma and platelet ratios early in resuscitation was associated with decreased mortality [50]. In 2015, an original investigation called the PROPPR Randomized Clinical Trial [51] examined data from patients with major bleeding after severe trauma receiving an early administration of red blood cells, plasma, and platelets in a 1:1:1 ratio compared with a 2:1:1 ratio. The authors found that there was no significant difference in mortality at 24 h and 30 days (these two ratios resulted therapeutically equivalent). In this study, 338 patients received blood products with a ratio of 1:1:1, and 342 patients received a ratio of 2:1:1 during resuscitation. Moreover, more patients treated with a 1:1:1 ratio achieved hemostasis and decreased death. On the contrary, no significant differences were reported with an increased use of platelets and plasma [48,49,50,51,52]. Stephens et al. [7] stated that the early use of a fixed ratio of red blood cells/platelets/plasma (1:1:1) needed further studies to be standardized in the treatment of post traumatic bleeding. Farrell et al. [53] leaned towards the idea of an early balanced replacement with blood components but confirmed that more research is required. To summarize, the best approach to transfusions is not known yet. Most studies supported the idea of a fixed balanced ratio of blood products to start early, while others preferred to use other tests, such as viscoelastic assays or scores of severities, before deciding about the ratio of transfusions [52,53,54]. 

Hospitals usually follow local protocols, and the most common accepted idea is that patients need to receive blood transfusions combined with other blood products in cases of traumatic bleeding, to restore the hemodynamic stability of patients, which is an important endpoint of resuscitation measures [27,53,54,55]. Protocolization and compliances with timely component ratios improves outcomes. Literature studies demonstrated a 2-fold difference in mortality rates between best- and worst-performing trauma centers. Finally, Thau et al. [56] emphasized the importance of molecular measurements in identifying the treatment-responsive “endotypes” of injured patients, as it is still unclear which patients with post-traumatic bleeding may benefit from a 1:1:1 (plasma/platelets/red blood cells) resuscitation strategy vs. 1:1:2 (plasma/platelets/red blood cells) ones [56] (Table 1). More studies are needed to explore this field.

## 7. European Guidelines for the Management of Post Traumatic Bleeding 

The current European guidelines [1] on the management of major traumatic bleeding and coagulopathy after a trauma recommend an immediate diagnosis and treatment of traumatic bleeding and coagulopathy to improve outcomes in severely injured patients. Pre-hospital administration of blood products has not yet received clear recommendation. It is important to assess the extent of traumatic bleeding through anatomical and physiological injury patterns, the mechanism of injury, and the patient’s response to initial support resuscitation. The authors recommend using the shock index score to determine the severity of hypovolemic shock and transfusion needs. Rossaint et al. [1] recommended that patients with a recognized source of bleeding and those with clinically present hemorrhagic shock require an immediate hemorrhage control measure. The repetition of the hemoglobin/hematocrit level could be a useful marker for bleeding. Blood lactate could monitor the extent of hemorrhage and tissue hypoperfusion. If the lactate measurement is not available, a base deficit may be a suitable alternative. Other tests include prothrombin time (PT)/international normalized ratio (INR), platelet count, and viscoelastic method. Patients on anticoagulants or antiplatelet therapy need to be strictly monitored. In the first phase of the trauma, a restricted volume replacement with a targeted systolic blood pressure of 80–90 mmHg should be maintained. If the hemorrhage is massive, crystalloids could be used in combination with vasopressors and, if needed, with colloid infusions to restore perfusion. Researchers have reported significant complications with large volumes of crystalloid, finding that infusions of ≥1.5 L of crystalloids were associated with increased mortality [50,51]. If blood transfusions are necessary, the target hemoglobin value should be 70–90 g/L. Red blood cell transfusions improve the volume status and restore arterial oxygen transport, which is decreased in cases of hemorrhagic shock. Red blood cell transfusions can be extensively used in patients admitted for trauma to replace blood loss and obtain bleeding control. In a recent study that analyzed the data of 21,433 patients, it was shown that the level of hemoglobin of 70–80 g/L did not result in more harm than the value of 90–100 g/L as a target to regulate transfusion support. Tranexamic acid is recommended in patients who are bleeding or at high risk of bleeding as soon as possible (within three hours after trauma at a dose of 1 g infused over 10 min, followed by an intravenous infusion of 1 g over eight hours) [57]. Other strategies to consider are supplementation with fibrinogen concentrate or cryoprecipitate, and red blood cells at a ratio of 1:2 and platelets if needed. It is crucial to implement guidelines for the management of bleeding in trauma patients locally and improve parameters for assessing and treating post-traumatic bleeding to achieve better patient outcomes.

## 8. Discussion

Acute hemorrhage due to trauma is a life-threatening condition that requires quick identification and treatment by emergency physicians to improve patients’ outcomes [58,59]. Coagulopathy is the other side of the coin. Regarding treatment for acute post-traumatic bleeding, literature research has considered standardized massive transfusion protocols as a key component to obtain better survival and a low rate of complications and organ failure. Every hospital trauma center usually has a massive transfusion protocol based on its resources and skills to facilitate diagnosis and treatment during a critical situation [60,61]. Emergency physicians should stratify the goals for patients’ resuscitation. First, it is of paramount importance to have an adequate damage control strategy [62,63], early transfusion and surgical techniques for blood saving. As regards transfusions, evidence have shown that the replacement with three main products packed red blood cells, fresh frozen plasma, and platelets with a ratio of 1:1:1 improved the survival of patients with traumatic bleeding and improved coagulopathy as well [58,64,65,66]. Other studies claimed similar effects in term of mortality with a ratio of 2:1:1. Few studies analyzed not a balanced ratio, for example, 1:2 (plasma/red blood cells) or 1:3 (plasma/red blood cells), or 1:1:4 (plasma/platelets/red blood cells) [67,68,69]. The administration of platelets, plasma, and cryoprecipitate has been associated with improved outcomes too, if administered early [70,71]. Regarding platelets, the use of cryopreserved platelets may be a feasible and tolerable alternative to fresh platelets in an emergency setting for traumatic patients. Fresh platelets have a short life with a risk of lacking availability [72].

An increase in survival has also been recorded with the use of whole blood (due to less volumes infused). For example, in a military setting, whole blood represents the preferred product for resuscitation of severe post-traumatic hemorrhage [55,64,65,66]. Whole blood group O can be safely transfused in patients of unknown blood group as emergent/initial support; this last contains low titers of antibodies anti-A and anti-B, can be stored for up to 35 days, and can supplement the blood components and coagulation factors that patients need. Furthermore, the transfusion of whole blood has a low risk of hemolysis and bacterial contamination and it is associated with a better survival of severely injured patients [67,68]. In addition, when needed, patients can receive rFVIIa (activated recombinant factor VII), 4-factor prothrombin complex concentrate (4F-PCC), and tranexamic acid, but studies have shown controversial results on patients’ survival [73,74,75]. 

The optimal transfusion strategy for traumatic bleeding is not well known. It is reasonable, as suggested by European guidelines, to start with a “ratio-driven” approach and move quickly to a “goal-directed” one (also known as the Copenhagen or hybrid approach). The decision to start with a balanced ratio of plasma/platelets/red blood cells unit or a partial ratio of 1:2 (plasma/red blood cells) is based on subjective clinical criteria and the availability of blood products by the blood bank [76,77,78,79]. The right time of transport, the use of blood tests [71] have not been standardized yet in the trials available right now [70,80]. It is a common opinion that the immediate recognition of patients who can benefit from immediate transfusions of blood products is essential. Many questions are still open for standard practice in the management of post-traumatic bleeding.

## 9. Conclusions

In conclusion, severe post-traumatic bleeding continues to be one of the leading causes of potentially preventable death in patients admitted to the emergency department after a traumatic injury. Emergency physicians should promptly manage patients with bleeding using a multidisciplinary approach, obtaining bleeding control with a personalized and tailored approach, and a balanced ratio of blood products to improve outcomes and save lives. Even though much progress has been made in the field of transfusions over the years, more research is required to add new insights. 

## Figures and Tables

**Figure 1 diagnostics-14-00708-f001:**
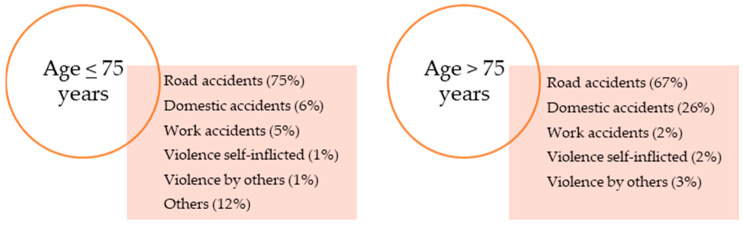
The most common causes of trauma, according to two age ranges [1,11].

**Figure 2 diagnostics-14-00708-f002:**
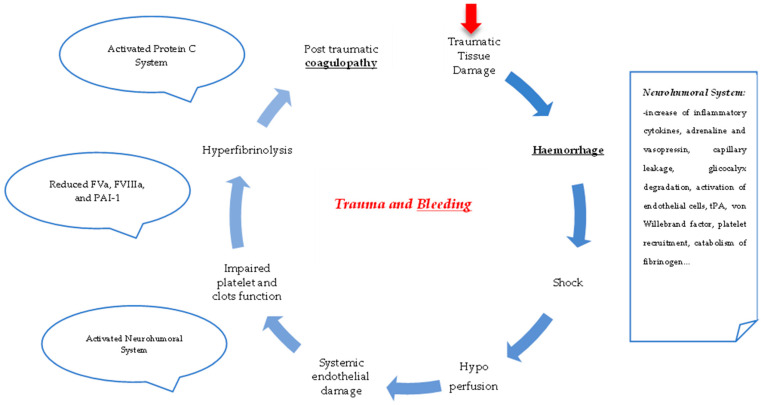
Traumatic bleeding and coagulopathy [1,16,17,18,20]. Abbreviations: tPA: tissue plasminogen activator; PAI-1: plasminogen tissue activator inhibitor type-1; FVa: activated factor V; FVIIIa: activated factor VIII.

**Figure 3 diagnostics-14-00708-f003:**
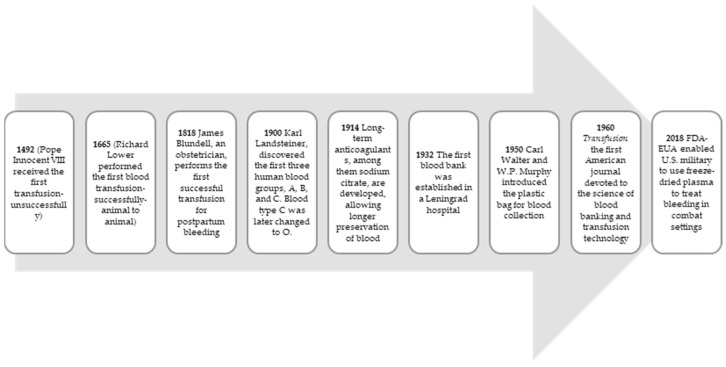
Some steps in the history of blood transfusions [31,32,33,34,35].

**Table 1 diagnostics-14-00708-t001:** Transfusions in injured patients.

Authors	Statement	Conclusion
Malone et al. J Trauma 2006 [38]	The prevention of coagulopathy is superior to its treatment. Ratio 1:1:1 (red blood cells/plasma/platelet) is easy to use and the relatively higher plasma and platelet doses seems to be associated with improved outcome.	The fixed volume ratios might allow the number and rate of administered units of red blood cells to be used for primary treatment of bleeding.
Terence O’Keeffe et al. Arch Surg 2008 [37]	A massive transfusion protocol offers a number of benefits. Massive transfusion protocol was defined as receiving a 1:1:1 ratio of red blood cells, plasma, and platelets after at least 4 units of transfused red blood cells within 1 h of emergency department presentation or 10 units within 24 h of ED arrival.	Benefits: decreased blood product transfusions, improved time to transfusion, and reduced blood bank and hospital charges.
John B. Holcomb et al. Jama 2015 [51]	Plasma, platelets, and red blood cells in a 1:1:1 ratio vs. a 1:1:2 ratio	No significant differences in mortality at 24 h or at 30 days.
Guerado et al. Eur J Trauma Emerg Surg 2016 [46]	Massive bleeding is an acute life-threatening event. Massive transfusion is the transfusion of ≥10 red blood units within 24 h, the transfusion of >4 red blood units in 1 h, the replacement of >50% of the total blood volume by blood products within 3 h. Massive transfusion can provoke further complications and problems.	The early diagnosis of the source of bleeding is essential. The primary treatment is the surgical control of the source of bleeding in combination with resuscitation measures.
Michael S. Farrell et al. Emergency Medicine Clinics of North America 2020 [53]	In injured patients, crystalloid should be limited and blood transfusions should be initiated early. Blood products should be utilized in a balanced ratio and should be guided by viscoelastic assays and endpoints of resuscitation.	The main point is to address the cause of the hemodynamic instability or coagulopathy in injured patients to maximize the benefit of transfusions.
Rossaint et al. Critical care 2023 [1]	Rapid control of bleeding; Coagulation support (fibrinogen concentrate or cryoprecipitate or red blood cells/fresh frozen plasma at a ratio of 1:1 or 2:1 as needed. A high platelet/red blood cells may be applied); Fresh frozen plasma guided by PT and/or APTT > 1.5; Fibrinogen supplementation if fibrinogen level < 1.5 g/L; Factor concentrated based on laboratory coagulation parameters and viscoelastic evidence of factor deficiency; Platelets to maintain platelets > 50 × 10^9^/L in patients with ongoing bleeding and >100 × 10^9^/L in patients with traumatic brain injury; Management of antithrombotic agents.	Immediate detection and management of traumatic coagulopathy improves outcomes of severely injured patients.
Matthew R. Thau et al. Jama 2023 [56]	Endotypes derived from plasma biomarkers in injured patients were associated with a differential response to (plasma, platelets and packed red blood cells) 1:1:1 vs. 1:1:2 resuscitation strategies.	The molecular heterogeneity in injured patients has implications for a tailored therapy for patients at high risk for adverse outcomes.

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
