# Peer review of "Blood Transfusion for Major Trauma in Emergency Department"

_diagnostics, 2024, doi:10.3390/diagnostics14070708_

Round 1

Reviewer 1 Report

Comments and Suggestions for Authors

Thank you for the opportunity to review this manuscript which attempts to review recent peer-reviewed literature regarding managing major trauma hemorrhage and coagulopathy with blood transfusion.

 1)      Reference 8 is not the best reference for general transfusion associated adverse events.  The reference use is specific for cancer patients.  A more general review reference, such as https://doi.org/10.1182/blood-2018-10-833988, doi: 10.1016/j.cll.2021.07.009. or https://doi.org/10.1016/j.mpsur.2022.05.025, Blood Safety Basics | CDC, or a WHO reference (https://iris.who.int/bitstream/handle/10665/42396/9241545399.pdf) would be more appropriate.

2)      Excess bleeding risk is not seen at INR >1.5.  INR is not the ideal measure for bleeding risk in general or for bleeding in trauma patients specifically since it was designed to be used to determine the effects of oral anticoagulants in non-bleeding patients.  INR >4.5 increases the risk for a major bleed. Normal warfarin target INR is between 2-3. In that range, the risk is related to bruising not dangerous bleeding. 

a.      INR >1.5 has been shown to be associated with an increased risk for adverse outcomes in a subset of trauma patients (https://doi.org/10.1097%2FCCM.0000000000000981).  Suggest the authors rewrite this section to state the fact with more precision.

b.      Studies (http://dx.doi.org/10.1097/TA.0b013e3182a9676c) have shown that INR is not an accurate predictor of coagulopathy of trauma, overestimating the coagulopathy in stable trauma.

3)      6. Definitions of Massive Transfusion and current protocols.  Appreciate the discussion about the lack of a practical day to day clinical definition of massive blood loss.  MBT abbreviation is given twice in back-to-back sentences, but not MTP or massive blood loss.  Several of the definitions given are not for massive blood transfusion as indicated in the manuscript, but for massive blood loss which may (or may not) require blood transfusions.  I suggest the authors rewrite this section to bring out this distinction better.  While MTP used to be defined by massive transfusions, increasingly with modern patient blood management practices where the right product is given at the right time and right dose, massive volumes of blood are not always given.

a.      Higher doses of yellow products (platelets, plasma, cryoprecipitate) was associated with improved outcomes only if given early.  There was a survivor bias in the old “Berne” concept where pRBCs were given first and only if the patient survived long enough, where they given yellow products. 

b.      As indicated in the previous point, the timing of yellow products is important.  Some of the trials focused on concurrent delivery of products, while others gave pRBCs first, with lagging plasma and platelets, and rarely cryoprecipitate.  Early access to yellow products is key, which is why the military prefers whole blood and why the civilian trauma surgeons in the US and elsewhere are pushing for pre-hospital access to whole blood.

4)      Other issues the authors may wish to incorporate into their manuscript

a.      Damage control resuscitation within the first hour (golden hour) or even better the platinum 15 minutes

b.      There is a 2-fold difference in mortality rates between best and worst-performing trauma centers - J Am Coll Surg 2009;209: 198–205, Injury  2013;44(12):1693-9

c.       Protocolization and compliances with timely component ratios improves outcomes - Cotton BA et al. J Trauma 2009, Injury 2015;46(1):21-8.  Average compliance in some institutions is 66% or worse, especially with use of platelets and cryoprecipitate when blood is given as components and not as whole blood.

d.      Increased plasma & platelet ratios early in resuscitation is associated with decreased mortality - Holcomb et al. JAMA Surg 2013 Feb;148(2):127-36

e.      PROPPR Trial (already mentioned) specifically mentions if component therapy is used, it should be transfused in ratios simulating whole blood.

f.        Limitation of use of crystalloid is summarized in Table 1 but not discussed in the body of the manuscript.  The only place where crystalloids are mentioned, the authors say it should be used with vasopressors and possibly colloids. Extensive use of crystalloids has exacerbated coagulopathy through dilution alone.  This should really be discussed more specifically.

5)      Discussion:

a.      This reviewer appreciates the authors pointing out that appropriate ratio-driven component therapy is a range of 1:1:1 or 1:1:2. The initial papers failed to demonstrate that 1:1:1 was superior to 1:1:2 so they should be considered as therapeutically equivalent.

b.      The authors should strongly consider rephrasing discussion from pointing to the need for massive transfusion protocol to the need for massive blood loss protocol.  I (again) point this out because early use of yellow products driven by algorithms or testing along with modern damage control principles and techniques often results in the net decrease of products used.  Delays in the onset of definitive treatment following injury, overuse of crystalloids, delays in blood component therapy including plasma/platelets/cryo (e.g. focusing on giving pRBCs only), waiting till the patient exhibits frank coagulopathy before giving yellow products are considered the reasons trauma patients required “massive” amounts of blood.

c.       The use of the Innerhofer reference (63) is a bit confusing to this reviewer.  The paper focused on fibrinogen levels following FFP (at least 15 ml/kg) versus FC or FibA10 dosing.  The authors, however, state that the authors conclude CFC’s (mainly FC) or as a single dose are always superior to plasma for reversal of trauma-induced coagulopathy.  The paper’s conclusions are a bit more nuanced.  It is easier to achieve adequate fibrinogen levels with FC or FibA10 dosing compared to use of FFP.  Concentrates are concentrated whereas FFP is not.  Since Fibrinogen is the first coagulation factor that reaches critical thresholds during trauma-induce consumption, it makes sense that FFP may not be able to “catch the patient up” without giving more that 15ml/kg and exposing the patient to large volumes of plasma. Suggest the authors modify the sentence to state something like…”CFC “may be” superior in reversing “serious” trauma-induced coagulopathy when FFP dosing of 15 ml/kg alone is likely unable to restore fibrinogen levels to normal levels.”

d.      Discussion about different centers using different ratios is confusing.  What is the point?  Towards the end of the Discussion section, the authors talk about availability of blood products as being a potential reason for partial ratios (often plasma and red cells only).  Are the authors suggesting access to a sufficient blood supply would lead to using a full ratio? Or does the relative short life of platelets (5-7 days) make it difficult to justify financially the stocking of platelets in EDs not used to seeing large volumes of massive trauma? This could be an interesting point about the need for platelets with longer shelf-life – e.g. cold-stored platelets or exploration into development of freeze-dried platelets or possibly more advanced therapies that could create a stable platelet product with longer shelf life.

Author Response

Thank you for the opportunity to review this manuscript which attempts to review recent peer-reviewed literature regarding managing major trauma hemorrhage and coagulopathy with blood transfusion.

  • Reference 8 is not the best reference for general transfusion associated adverse events.  The reference use is specific for cancer patients.  A more general review reference, such as https://doi.org/10.1182/blood-2018-10-833988, doi: 10.1016/j.cll.2021.07.009. or https://doi.org/10.1016/j.mpsur.2022.05.025, Blood Safety Basics | CDC, or a WHO reference (https://iris.who.int/bitstream/handle/10665/42396/9241545399.pdf) would be more appropriate.

We modified the reference n. 8

Excess bleeding risk is not seen at INR >1.5.  INR is not the ideal measure for bleeding risk in general or for bleeding in trauma patients specifically since it was designed to be used to determine the effects of oral anticoagulants in non-bleeding patients.  INR >4.5 increases the risk for a major bleed. Normal warfarin target INR is between 2-3. In that range, the risk is related to bruising not dangerous bleeding.  INR >1.5 has been shown to be associated with an increased risk for adverse outcomes in a subset of trauma patients (https://doi.org/10.1097%2FCCM.0000000000000981).  Suggest the authors rewrite this section to state the fact with more precision.  Studies (http://dx.doi.org/10.1097/TA.0b013e3182a9676c) have shown that INR is not an accurate predictor of coagulopathy of trauma, overestimating the coagulopathy in stable trauma.

Thank you for your observation, we rewrite the sentence

  • Definitions of Massive Transfusion and current protocols.  Appreciate the discussion about the lack of a practical day to day clinical definition of massive blood loss.  MBT abbreviation is given twice in back-to-back sentences, but not MTP or massive blood loss.  Several of the definitions given are not for massive blood transfusion as indicated in the manuscript, but for massive blood loss which may (or may not) require blood transfusions.  I suggest the authors rewrite this section to bring out this distinction better.  While MTP used to be defined by massive transfusions, increasingly with modern patient blood management practices where the right product is given at the right time and right dose, massive volumes of blood are not always given.
  1. Higher doses of yellow products (platelets, plasma, cryoprecipitate) was associated with improved outcomes only if given early.  There was a survivor bias in the old “Berne” concept where pRBCs were given first and only if the patient survived long enough, where they given yellow products.  
  2. As indicated in the previous point, the timing of yellow products is important.  Some of the trials focused on concurrent delivery of products, while others gave pRBCs first, with lagging plasma and platelets, and rarely cryoprecipitate.  Early access to yellow products is key, which is why the military prefers whole blood and why the civilian trauma surgeons in the US and elsewhere are pushing for pre-hospital access to whole blood.

We added these two points as suggested

4)      Other issues the authors may wish to incorporate into their manuscript

  1. Damage control resuscitation within the first hour (golden hour) or even better the platinum 15 minutes
  2. There is a 2-fold difference in mortality rates between best and worst-performing trauma centers - J Am Coll Surg 2009;209: 198–205, Injury  2013;44(12):1693-9
  3. Protocolization and compliances with timely component ratios improves outcomes - Cotton BA et al. J Trauma 2009, Injury 2015;46(1):21-8.  Average compliance in some institutions is 66% or worse, especially with use of platelets and cryoprecipitate when blood is given as components and not as whole blood.
  4. Increased plasma & platelet ratios early in resuscitation is associated with decreased mortality - Holcomb et al. JAMA Surg 2013 Feb;148(2):127-36
  5. PROPPR Trial (already mentioned) specifically mentions if component therapy is used, it should be transfused in ratios simulating whole blood.
  6. Limitation of use of crystalloid is summarized in Table 1 but not discussed in the body of the manuscript.  The only place where crystalloids are mentioned, the authors say it should be used with vasopressors and possibly colloids. Extensive use of crystalloids has exacerbated coagulopathy through dilution alone.  This should really be discussed more specifically.

Thank you so much, we incorporated these issues in the manuscript

5)      Discussion:

  1. This reviewer appreciates the authors pointing out that appropriate ratio-driven component therapy is a range of 1:1:1 or 1:1:2. The initial papers failed to demonstrate that 1:1:1 was superior to 1:1:2 so they should be considered as therapeutically equivalent.
  2. The authors should strongly consider rephrasing discussion from pointing to the need for massive transfusion protocol to the need for massive blood loss protocol.  I (again) point this out because early use of yellow products driven by algorithms or testing along with modern damage control principles and techniques often results in the net decrease of products used.  Delays in the onset of definitive treatment following injury, overuse of crystalloids, delays in blood component therapy including plasma/platelets/cryo (e.g. focusing on giving pRBCs only), waiting till the patient exhibits frank coagulopathy before giving yellow products are considered the reasons trauma patients required “massive” amounts of blood. 
  3. The use of the Innerhofer reference (63) is a bit confusing to this reviewer.  The paper focused on fibrinogen levels following FFP (at least 15 ml/kg) versus FC or FibA10 dosing.  The authors, however, state that the authors conclude CFC’s (mainly FC) or as a single dose are always superior to plasma for reversal of trauma-induced coagulopathy.  The paper’s conclusions are a bit more nuanced.  It is easier to achieve adequate fibrinogen levels with FC or FibA10 dosing compared to use of FFP.  Concentrates are concentrated whereas FFP is not.  Since Fibrinogen is the first coagulation factor that reaches critical thresholds during trauma-induce consumption, it makes sense that FFP may not be able to “catch the patient up” without giving more that 15ml/kg and exposing the patient to large volumes of plasma. Suggest the authors modify the sentence to state something like…”CFC “may be” superior in reversing “serious” trauma-induced coagulopathy when FFP dosing of 15 ml/kg alone is likely unable to restore fibrinogen levels to normal levels.” 

We modified and removed

  1. Discussion about different centers using different ratios is confusing.  What is the point?  Towards the end of the Discussion section, the authors talk about availability of blood products as being a potential reason for partial ratios (often plasma and red cells only).  Are the authors suggesting access to a sufficient blood supply would lead to using a full ratio? Or does the relative short life of platelets (5-7 days) make it difficult to justify financially the stocking of platelets in EDs not used to seeing large volumes of massive trauma? This could be an interesting point about the need for platelets with longer shelf-life – e.g. cold-stored platelets or exploration into development of freeze-dried platelets or possibly more advanced therapies that could create a stable platelet product with longer shelf life.

As regards points a),b),c),d) we modified discussion

Reviewer 2 Report

Comments and Suggestions for Authors

Thank you for the opportunity to read this interesting manuscript. Please take into account the following comments:

1) the methodology is not clear. There is definitely a lack of an appropriate diagram (I recommend: http://www.prisma-statement.org/documents/PRISMA%202009%20flow%20diagram.pdf ) where the path for selecting the final set of references will be described.

2) no references in the text to individual FIGURES

3) why were only 7 publications selected for table 1? What did the authors suggest? There is no more detailed list (in a table) of selected studies

4) the final conclusions do not apply at all to blood transfusions in ED conditions

5) I have the impression that the authors devoted too much space to describing the mechanisms of injury and bleeding complications, instead of focusing on the techniques and frequency of blood transfusion in the ED.

6) it is worth the authors mentioning priority life-saving treatments in the case of post-traumatic hemorrhages in their work. I propose to add recommendations for the use of tourniquets to the literature: Leszczyński P, Charuta A, Zacharuk T. Cadaver as an educational tool increasing the effectiveness of Combat Application Tourniquet use in extremity injuries. TURKISH JOURNAL OF TRAUMA AND EMERGENCY SURGERY, 2021; 27(2): 161-166.

DOI: 10.14744/tjtes.2020.35737

7) There is also no description of exemplary tests and scales (e.g. MEW) in predicting the need for blood transfusion - please take into account the exemplary publication: Krishna BSG, Goud DPK, Velavarthipati RS, Priya PS, Harish KM, Praveen K. Comparison of Glasgow Blatchford, pre -endoscopic Rockall, and modified early warning score systems to predict the clinical outcome of patients with upper gastrointestinal bleeding in the emergency. Crit. Care Innov. 2023; 6(3): 37-51. DOI: 10.32114/CCI.2023.6.3.37.51

Author Response

Thank you for the opportunity to read this interesting manuscript. Please take into account the following comments:

1) the methodology is not clear. There is definitely a lack of an appropriate diagram (I recommend: http://www.prisma-statement.org/documents/PRISMA%202009%20flow%20diagram.pdf ) where the path for selecting the final set of references will be described.

 This is a narrative review, not a systematic review. We added PICOST 

  • no references in the text to individual FIGURES

We introduced in the text

  • why were only 7 publications selected for table 1? What did the authors suggest? There is no more detailed list (in a table) of selected studies

We represented a timeline of main studies

4) the final conclusions do not apply at all to blood transfusions in ED conditions

 We modified conclusions

5) I have the impression that the authors devoted too much space to describing the mechanisms of injury and bleeding complications, instead of focusing on the techniques and frequency of blood transfusion in the ED.

 We modified the manuscript

6) it is worth the authors mentioning priority life-saving treatments in the case of post-traumatic hemorrhages in their work. I propose to add recommendations for the use of tourniquets to the literature: Leszczyński P, Charuta A, Zacharuk T. Cadaver as an educational tool increasing the effectiveness of Combat Application Tourniquet use in extremity injuries. TURKISH JOURNAL OF TRAUMA AND EMERGENCY SURGERY, 2021; 27(2): 161-166. DOI: 10.14744/tjtes.2020.35737

We added it

 7) There is also no description of exemplary tests and scales (e.g. MEW) in predicting the need for blood transfusion - please take into account the exemplary publication: Krishna BSG, Goud DPK, Velavarthipati RS, Priya PS, Harish KM, Praveen K. Comparison of Glasgow Blatchford, pre -endoscopic Rockall, and modified early warning score systems to predict the clinical outcome of patients with upper gastrointestinal bleeding in the emergency. Crit. Care Innov. 2023; 6(3): 37-51. DOI: 10.32114/CCI.2023.6.3.37.51

We have described some scales with parameters used

Reviewer 3 Report

Comments and Suggestions for Authors

Dear authors,

The review at hand is a nice summary of evidence on trauma and transfusion in the ED. However, there are some flaws that need to be corrected before it becoming eligible for publication:

-) Methods: This is too little information. Please read other narrative reviews to see what information is required here; for instance at least all details of your search strategy (when developed, who developed it, when applied,...), your PICOST, your PRISMA flowchart or similar, etc.

-) Page 2, line 84: There is no "the Delphi study", a Delphi study is a study type!

-) All figures must be refered to in the text, and cited appropriately. For instance, where does the information in Figure 1 come from? Etc.

-) Table 1 is quite confusing: What do you mean by "Main points of transfusions"? What exactly is depicted here?

-) The discussion is rather long and unstructured; it would benefit from sub-headings and a clearer train of thought.

-) Your conclusion is somewhat not on point - what are really the learning points from your review? Surely not that trauma is a leading cause of death and that future research should be undertaken (this was already clear beforehand).

Comments on the Quality of English Language

Moderate style adaptations necessary.

Author Response

Dear authors,

The review at hand is a nice summary of evidence on trauma and transfusion in the ED. However, there are some flaws that need to be corrected before it becoming eligible for publication:

-) Methods: This is too little information. Please read other narrative reviews to see what information is required here; for instance at least all details of your search strategy (when developed, who developed it, when applied,...), your PICOST, your PRISMA flowchart or similar, etc.

We introduced PICOST, this is a narrative review

-) Page 2, line 84: There is no "the Delphi study", a Delphi study is a study type!

Thank you for your observation, we corrected

-) All figures must be refered to in the text, and cited appropriately. For instance, where does the information in Figure 1 come from? Etc.

We modified the manuscript

-) Table 1 is quite confusing: What do you mean by "Main points of transfusions"? What exactly is depicted here?

We modified the title

-) The discussion is rather long and unstructured; it would benefit from sub-headings and a clearer train of thought.

We corrected the discussion

-) Your conclusion is somewhat not on point - what are really the learning points from your review? Surely not that trauma is a leading cause of death and that future research should be undertaken (this was already clear beforehand).

We modified

Comments on the Quality of English Language. Moderate style adaptations necessary.

We checked the manuscript and corrected

Round 2

Reviewer 1 Report

Comments and Suggestions for Authors

All issues addressed.  Thank you.

Author Response

Thank you so much 

Reviewer 2 Report

Comments and Suggestions for Authors

accept

Author Response

Thank you so much 

Reviewer 3 Report

Comments and Suggestions for Authors

Dear authors,

Thank you for having addressed my comments. However, there are still some issues:

-) It is "PICOST" (the S is missing in your text)

-) Your figures do not have any citations or further explaining in their legends.

Comments on the Quality of English Language

Moderate spelling and grammar mistakes.

Author Response

 It is "PICOST" (the S is missing in your text)

Yes, we corrected 

-) Your figures do not have any citations or further explaining in their legends.

Thank you for your observation, we corrected, we added references and explained abbreviations    Moderate spelling and grammar mistakes. We revised the manuscript and corrected mistakes   

Round 3

Reviewer 3 Report

Comments and Suggestions for Authors

None.